# Geographical Disparities and Settlement Factors and Mental Health of Refugees Living in Germany

**DOI:** 10.3390/ijerph20054409

**Published:** 2023-03-01

**Authors:** Julian Grabo, Gerard Leavey

**Affiliations:** 1Global Health, Maastricht University, 6221 LK Maastricht, The Netherlands; 2Bamford Centre for Mental Health & Wellbeing, Ulster University, Coleraine BT51 5SA, UK

**Keywords:** refugees, mental health, discrimination

## Abstract

(1) Background: Approximately half of all refugees living in Germany experience discrimination, which may negatively affect their mental health. Moreover, German refugees have experienced hostility, especially in eastern regions. (2) Aims: We examined the effect of perceived discrimination on refugees’ mental health in Germany, with a particular focus on possible regional differences of refugee mental health and perceived discrimination. (3) Method: The data of 2075 refugees who arrived in Germany between 2013 and 2016, from a large-scale survey, was analysed using binary logistic regression. The refugee health screener, 13-item version, was used to assess psychological distress. All effects were investigated for the entire sample and both sexes independently. (4) Results: A third of refugees experienced discrimination which increased the risk of psychological distress (*OR* = 2.25 [1.80, 2.8]). Those living in eastern Germany were more than twice as likely to report experiences of discrimination, compared to their counterparts living in western Germany (*OR* = 2.52 [1.98, 3.21]). Differences were noted between males and females, and religious attendance. (5) Conclusions: Perceived discrimination is a risk factor for refugee mental health, particularly female refugees in eastern Germany. An east–west regional difference may be explained by socio-structural factors, rural placement, differential historical exposure to migrant populations, and a greater presence of right-wing and populist parties in eastern Germany.

## 1. Introduction

Over the last decade, the number of refugees worldwide has increased, to 26 million in 2019 [1]. This trend is also visible in Germany, which received 2.1 million new asylum applications in the last decade, the highest number in the world [1]. Most are from Syria, Afghanistan, Iraq, Eritrea, and Nigeria, due to war, violence, and persecution, amongst other reasons [1,2]. The mental health of refugees has been highlighted as a challenge for services [3,4]. High rates of post-traumatic stress disorder (PTSD), major depressive disorder, and anxiety disorders have been noted among refugee populations [5,6]. These problems may persist for several years after resettlement [7]. However, the considerable variation in the epidemiology of refugee psychiatric disorders, highlights differences in populations’ pre- and post-settlement experiences, and methodological differences in research [6,7,8,9,10,11]. The ecological model of refugee distress, by Miller and Rasmussen [12], explains the high prevalence of mental health problems in refugees, due to potentially traumatic pre-migratory experiences, migration experiences, and post-migratory stressors [13]. Pre-migration experiences include, among others, war exposure, torture, prosecution, and displacement. They have been extensively linked to the development of PTSD and other mental disorders [7,9].

Following displacement, refugees may experience numerous stressors in their daily lives, such as poverty, unemployment, family separation, uncertainty regarding their asylum status, discrimination, loss of social support networks, and more [12]. These factors are often compounded by the experience of daily stressors, with exacerbating negative effects [14]. For example, limited employment opportunities restrict access to stable housing, causing greater instability in an individual’s life, which may lead to increased loneliness [14]. Protective factors against mental health problems include, having an occupation and social support [15,16]. Overall, greater exposure to pre- and post-migratory stressors and risk factors are predictive of worse mental health [7,15].

Perceived discrimination has been found to negatively affect the mental health and well-being of refugees [15,17,18,19,20,21,22,23,24]. Other daily stressors include, finding housing and employment, access to healthcare [25], as well as reinforcing social exclusion [26]. Importantly, a recent study found that half of all refugees living in Germany experience discrimination [27]. Similar levels of refugee discrimination have been reported in other countries, such as the USA [28], Australia [29], and Denmark [30]. Refugee and asylum-seeking populations are often placed into already disadvantaged neighbourhoods, where racism and xenophobic tendencies are common. Thus, the placement and settlement policies of various European countries have been criticised for failing to take account of local economic and political pressures.

### Refugees in Eastern and Western Germany

During the unprecedented arrival of refugees into the European Union in 2015 and 2016, and the resulting record number of refugees living in Germany, anti-migrant movements, like the Alternative für Deutschland (AFD) (AFD translates to Alternative for Germany) and the Patriotische Europäer gegen die Islamisierung des Abendlandes (PEGIDA) (PEGIDA translates to Patriotic Europeans Against the Islamisation of the Occident), gained popularity, and attacks on refugees became more frequent [31]. Anti-immigrant sentiments appeared to be more frequent in eastern Germany [32,33,34,35,36,37,38,39,40]. It is important to note that these differences can be attributed to multiple socio-structural factors, by which German federal states differ systematically [31,41]. However, other recent research has begun to challenge the notion of difference in attitudes towards refugees between eastern and western Germany [42,43], possibly explained by post-war political and social divergence, and societal exposure to inward migrant and refugee populations, resulting in what has been termed ‘migrant scepticism” in some European regions such as eastern Germany [44].

We aimed to investigate the effect of discrimination on the mental health of refugees living in Germany, and potential eastern–western difference in the prevalence of perceived discrimination in this population. We hypothesised that experience of discrimination is positively associated with psychological distress, and that refugees in eastern Germany would report experiencing higher levels of discrimination compared to those in western Germany.

## 2. Materials and Methods

This cross-sectional correlational study used secondary data analysis of data from the second wave (2017) of the IAB-BAMF-SOEP survey of refugees in Germany. This annual, representative household survey includes refugees, and all other individuals living within the household. The sample was drawn from the Ausländerzentralregister (Ausländerzentralregister translates to Central Register of Aliens) of the Bundesamt für Migration und Flüchtlinge (BAMF) (BAMF translates to Federal Office for Migration and Refugees), based on 169 randomly selected geographic sample points [45]. Participation was voluntary, and if possible, all close family members, including parents, siblings, children, and partners living in one household with the selected individual were interviewed. From the IAB-BAMF-SOEP survey refugees, asylum seekers, individuals with protection status, and rejected asylum seekers with a temporary resident’s permit, who are at least 18 years old, and arrived in Germany between 1 January 2013 and 31 January 2016, were included within this study.

### 2.1. Variables and Measurement Instruments

#### 2.1.1. Outcome Variables

Psychological distress was assessed through the 13-item version of the Refugee Health Screener (RHS-13) [46]. The original 15-item version was developed for refugee populations, based on symptoms of depression, anxiety, and post-traumatic stress [46], with excellent internal consistency (Cronbach’s α = 0.92) [47]. The shorter RHS-13 also showed high internal consistency (Cronbach’s α = 0.96) [47,48]. Each item was scored on a five-point Likert scale (0–4; total range of scores 0–52), with the points being labelled: ‘not at all’ (0), ’a little bit’ (1), ‘moderately’ (2), ‘quite a bit’ (3), ‘extremely’ (4). These items were summed, with a score of 11 or higher being indicative of psychological distress [47].

Perceived discrimination was assessed through a single-item measure, asking whether the individual has experienced being treated disadvantageously due to their origin in the past two years (coded yes or no).

Sociodemographic variables were age, sex, religion, country of origin, marital status residence status (includes recognised refugees, asylum seeker, and others with protection status, temporary suspension of deportation, in proceedings, and other), and length of stay in Germany.

German language proficiency was measured through three items (conversational, reading, and writing) on a scale from 1 (very well) to 5 (not at all). The average of the three scores was used to indicate German language skills on the same scale from 1 to 5.

Level of education was based on the International Standard Classification of Education as follows: low (early childhood education, primary education, lower secondary education), medium (upper secondary, post-secondary non-tertiary education, short-cycle tertiary education), and high (bachelor’s or master’s degree or equivalent, doctoral or equivalent degree) education [49].

#### 2.1.2. Other Covariates of Interest

The variable eastern/western Germany indicated in which region of Germany a refugee lived. The federal states Brandenburg, Mecklenburg-Vorpommern, Saxony, Saxony-Anhalt, Berlin, and Thuringia were included in eastern Germany. Whereas the federal states Hesse, Bavaria, Baden-Württemberg, Hamburg, Lower-Saxony, Rhineland-Palatinate, Saarland, Bremen, North Rhine-Westernphalia, and Schleswig-Holstein were included in western Germany. The distinction rural/urban was made, based on the definition of urban areas by the Bundesinstitut für Bau-, Stadt- und Raumforschung (BBSR translates to Federal Institute for Research on Building, Urban Affairs, and Spatial Development) (BBSR) [50]. Agglomeration areas and urbanised areas were jointly considered as urban areas [50].

Negative migration events were, an experience of one or more negative events on the migration journey to Germany (financial fraud, financial exploitation, sexual harassment, physical attacks, shipwreck, robbery, blackmail, or imprisonment).

Social support was operationalised as how many important people and close confidants an individual had in different areas of their life. Individuals were able to name up to five individuals. Additionally, individuals were asked whether they attend religious events or meetings, in the variable religious events/meetings. The variable has the option yes and never.

#### 2.1.3. Analysis

The data were analysed using IBM SPSS statistics 25. Due to the household sampling method used in the IAB-BAMF-SOEP survey, multiple individuals from the same households were included in the sample. Since the observations from the same household correlated based on the household, this violates the assumption of independence of errors [51]. To address this issue, all households from which more than one individual was interviewed were identified, and one individual was randomly selected from each household. Cases with missing values were excluded pairwise. This means that for all procedures, all observations available without any missing data were used. Descriptive statistics were calculated as means and standard deviation for continuous variables, and relative and absolute frequencies were employed for categorical variables. Logistic regression was used to assess associations between psychological distress and other variables. The same procedure was followed for perceived discrimination as the outcome variable.

The associations between psychological distress, living in eastern or western Germany, and perceived discrimination were investigated using univariate binary logistic regression. All regression models were corrected for demographic factors, and other covariates that may have influenced any associations between the variables of interest. Additionally, logistic regression was conducted for males and females separately, to investigate whether the associations observed within the total models held independently for both males and females. All variables’ odds ratios (*OR*) and 95% confidence intervals (95% CI) were reported. A sensitivity analysis of the results was conducted by randomly drawing a second random sample of individuals from households with more than one individual participating, and repeating the analysis. Individuals from a household including multiple participants that were included in the main analysis could not be selected again. It was checked whether each association remained the same in the sensitivity analysis. German proficiency was assessed through a 1–5 scale, which was computed through the mean of three items: conversational, reading, and writing. Age was transformed into a categorical variable with four approximately equally sized categories (18–27; 28–34; 35–41; and 42+ years).

## 3. Results

Due to the drawing of a random sample of one individual per household, 499 of the initial 2574 cases were excluded in the main analysis. The final sample included 2075 individuals. The internal reliability of the RHS-13 was excellent (Cronbach’s α = 0.91). The sample demographics (Table 1) show that the sample consisted mostly of young married Muslim men, from the Middle Eastern and the Horn of Africa, with a low level of education. Most refugees lived in western Germany (80%, *n* = 1658) and in urban areas (69%, *n* = 1425). Most individuals had their protection status awarded (75%, *n* = 1425). However, 16% (*n* = 325) of the participants’ decisions concerning their residence status were still in progress, 5% (*n* = 95) had their deportation suspended, and 4% (*n* = 88) of cases had another residency status. Just over half the sample (51%) attended religious events or meetings. The length of stay in Germany was 2.4 years (*SD* = 0.73). Most people had two close confidants (*M* = 2.15, *SD* = 1.49). The mean number of individuals who supported a refugee’s progress was 1.63 (*SD* = 1.45), while the mean number of individuals who could tell the refugee unpleasant truths was 1.42 (*SD* = 1.42). Lastly, 38% (*n* = 757) had experienced one or more negative experiences during their journey to Germany.

### 3.1. Psychological Distress among Refugees in Germany

Adjusted for other factors, discrimination was associated with psychological distress (Table 2). Refugees living in eastern Germany were twice as likely to have psychological problems than those in western Germany, while being a woman increased the odds of psychological distress by two-thirds. The country of origin significantly explained the variation within psychological distress. However, only the odds of psychological distress of Afghan refugees differed significantly from Syrian refugees, with their odds being 85% higher.

Older refugees (42 years+) had a higher risk of psychological distress compared to their younger counterparts; increased risk of psychological distress was found for the experience of negative events and lower German language skills (29% for every lower point on the scale).

Lastly, the results suggest that every additional close confidant that can tell a refugee an unpleasant truth about themselves increases the refugee’s odds of psychological distress by 13%. Factors excluded in the final model were the number of people that support the refugee in their professional progress, the number of close confidants with whom the participant can share their thoughts and feelings, religion, education, and urban/rural residence.

### 3.2. Psychological Distress by Sex

Stratifying by sex, the association between psychological distress and perceived discrimination holds for both sexes, while the association between eastern and western Germany and psychological distress remained significant for females only (*p* = 0.002). Women living in eastern Germany had twice the odds of psychological distress compared to those living in western Germany (*OR* = 2.09 CI = 1.32, 3.33). A smaller, non-significant trend was noted for males (*OR* = 1.11 CI = 0.80–1.54). A significant association between age and psychological distress was found for males and females, but with less clarity for males. Older women had an increased risk of psychological distress compared to those aged 18–27 years (*OR* = 6.06 CI = 3.40–10.78). A significant association between country of origin and psychological distress was found in the male sample (*p* = 0.003), but not in the female sample (*p* = 0.18). Yet, the odds of refugees from Afghanistan presenting with psychological distress remained significantly increased compared to refugees from Syria in the female sample (*OR* = 1.98 [1.11, 3.51]). German language skills were positively associated with psychological distress in the male sample (*p* < 0.001), but failed to reach significance for females (respectively, *OR* = 1.39 [1.19, 1.62]; and *OR* = 1.20 [0.99, 1.45]).

A significant association between psychological distress and the number of close confidants that can tell an unpleasant truth to the participant was found for males (*p* = 0.02) and females (*p* = 0.02). For every additional confidant, the odds of psychological distress were found to increase by 11% for males (*OR* = 1.11 [1.01, 1.22]) and 17% for females (*OR* = 1.17 [1.03, 1.33]). Moreover, the experience of one or more negative migration events was only significantly associated with psychological distress for males (*p* < 0.001), and not for females (*p* = 0.44). The odds of males that experienced such an event were increased by 76% compared to those who did not experience any such event (*OR* = 1.76 [1.33, 1.28]).

### 3.3. Sensitivity Analysis

The findings presented above, of psychological distress among refugees who came to Germany between 2013 and 2016, were replicated in a sensitivity analysis. All associations reported above remained the same in the sensitivity analysis, except for the association between German language skills and psychological distress in the female sample. This association was significant (*p* = 0.018), indicating that worse German skills are connected to an increase in the probability of psychological distress in female refugees (*OR* = 1.28 [1.04, 1.56]).

### 3.4. Perceived Discrimination of Refugees in Germany

We examined eastern–western differences in refugee’s perceived discrimination, using logistic regression (Table 3). The odds of discrimination were 2.5 times higher for refugees living in eastern Germany compared to refugees living in western Germany. Moreover, those living in rural areas were more likely to report discrimination than those living in urban areas. Moreover, we noted that perceived discrimination decreased by 6.5% for every year a person aged, and only Iraqi refugees had a significantly lower risk of being discriminated against compared to Syrians. Lastly, attendance at religious events or meetings increased the risk of discrimination by 28%.

Other variables were investigated with regards to perceived discrimination, but were not included in the final model, since no significant association with perceived discrimination was found. These included religious denotation, residency status, level of education, and marital status.

### 3.5. Perceived Discrimination by Sex

Two separate logistic regressions, including the same variables as the analysis above, were performed for males and females, to investigate differences in the factors predicting perceived discrimination between the sexes. N.B., 1308 individuals were included in the male analysis (62 cases excluded due to missing data). The female model was constructed by analysing 685 observations (20 were excluded due to missing data).

The association between living in eastern or western Germany and perceived discrimination was significant for both sexes (*p* < 0.001) (male: *OR* = 2.67 [1.98, 3.61] and female: *OR* = 2.35 [1.55, 3.55]). The association between rural/urban resettlement areas and perceived discrimination did not hold for males (*p* = 0.610), but was significant for females (*p* = 0.002). Female refugees living in rural areas were more at risk of being discriminated against than their counterparts living in urban areas (*OR* = 1.81 [1.25, 2.61]). Moreover, age was significantly associated with perceived discrimination among males *(p* < 0.001), but not females. The association indicates that males report that they are less discriminated against the older they become (*OR* = 0.97 [0.96, 0.98]).

Country of origin remained significant for males and females. Furthermore, Iraqi male and female refugees had significantly lower odds of perceived discrimination than Syrian male and female refugees (male: *OR* = 0.65 [0.42, 0.99], female: *OR* = 0.35 [0.17, 0.69]). Attending religious events was associated with perceived discrimination for males *(p* = 0.005), but not females (*p* = 0.901). Male refugees attending religious events or meetings were one and a half times more likely to report perceived discrimination than non-attenders (*OR* = 1.44 [1.11, 1.85]). However, whether female refugees attended religious events did not affect their odds of being discriminated against (*OR* = 1.02 [0.71, 1.47]).

### 3.6. Sensitivity Analysis

Except for the association between living in rural or urban areas and perceived discrimination, all associations were replicated by the sensitivity analysis.

## 4. Discussion

Refugees have high rates of mental health illnesses, especially PTSD, depressive disorders, and anxiety disorders [5], due to the combined impact of traumatic pre-migratory experiences and post-migratory stressors. We investigated the effect of perceived discrimination on the mental health of refugees who came to Germany between 2013 and 2016. Additionally, it was examined how different factors, especially living in eastern or western Germany, affect the prevalence of perceived discrimination of refugees. As expected, refugees who reported being discriminated against were found to have worst mental health. Regardless of sex, refugees were found to have a higher chance of being discriminated against in the eastern part of Germany, compared to western Germany. This accords with evidence of violence against refugees in eastern Germany during 2015–2016 [34], and more xenophobic attitudes in this population [37]. This may be explained by systematically differing socio-structural factors between eastern and western Germany, including the size of migrant populations [31] and right-wing and populist parties being more acceptable in eastern Germany [35]. However, the impact of living in eastern Germany appears to increase the vulnerability to psychological distress for female refugees only, possibly indicating that some of these factors may disproportionately affect female refugees and their mental health, in eastern Germany. Many female refugees originate from the Middle East [52], where headscarves are worn for religious and cultural reasons. However, headscarves may provide a greater level of ‘ethnic visibility’ for women than men, leaving them more vulnerable to racist behaviour within Germany [53,54]. Historically and currently, fewer foreigners lived in eastern Germany than western Germany [52], with less exposure to refugees and migrant communities [37]. As in bordering countries such as Hungary, Poland, and the Czech Republic, which all strongly rejected any welcome to refugees, eastern Germans have also been more resistant to the European Union’s ‘open-door’ policies. This resistance has also been expressed in the political system, with the far-right Alternative for Germany (AfD) Party winning 13% of the national vote in the 2017 election, but recording double that percentage in Saxony (eastern Germany). The higher levels of anti-migrant/refugee sentiment in eastern Germany may be explained by Allport’s contact theory [55], which holds that under certain conditions, stereotyping, prejudice, and discrimination could be mitigated by greater interpersonal contact between groups. However, compared to their western counterparts, few people growing up in Soviet East Germany, prior to unification, were able to watch western television or listen to radio broadcasts, limiting multicultural exposure. Additionally, foreigners in East Germany, from Cuba and African countries, were mostly visitors rather than settlers, unlike the Turkish in West Germany. Scarce resources, actual or perceived, may also determine hostility to foreigners. Since unification, the unemployment rate in the east is 13%, remaining almost double that in the west (Federal Statistical Office Germany, 2010).

As noted in sociological studies, the levels of prejudice against both Muslims and immigrants are consistently higher in eastern European and former Soviet Bloc countries than in western Europe, even though the determinants are similar. In this, other social scientists [56], while pointing to the legacy of Communism, also highlight the economic determinants as being more important, i.e., the higher the poverty, the greater the antipathy towards migrants.

The prevalence of psychological distress is different for people from different countries. Males and females from Afghanistan appear to be at an increased risk, possibly related to war and conflict during the last four decades [57]. Iranian refugees experienced less discrimination than Syrian refugees, who make up the majority of refugees in Germany [52]. We also found a negative association between attending religious meetings or events and being discriminated against, for men but not women. Most migrants and refugees in Germany are Muslims [52], and anti-Muslim prejudice in Germany [58] may impact more on men, since they are more likely to attend mosque than women. Importantly, women from traditional Muslim communities tend to have a more restrictive social existence, and are less likely to be economically active; consequently, women are less socially visible and at risk of exposure to discriminatory behaviour and racial violence. However, some European states’ policies against the wearing of traditional dress may lead to even greater invisibility among religious women. Thus, there is evidence that veiled Muslim women face discrimination in Germany and the Netherlands, particularly when applying for jobs that require a high level of customer contact [59]. Moreover, we found associations between male refugees who experienced traumatic events during the migration journey and psychological distress. [60]. In Germany, most refugees are male, while most residency permits awarded for family reasons are given to women whose husbands were awarded protection status [52]. Therefore, many female refugees in the sample may not have fled their country with their husbands, but arrived in Germany through family reunification afterwards, avoiding potentially traumatic events during their journey [61].

Social support was inconsistent, with one of the three items showing an effect with psychological distress. Paradoxically, perhaps, those who reported having close confidants were more likely to have psychological distress. However, this may reflect something of the cultural differences, and challenging social contexts, such as greater exposure to shame and/or higher level of dependants with support needs (rather than providing support) [62]. Importantly, we also noted that perceived discrimination decreases with age. While we cannot firmly conclude that this equates with length of stay, it is certainly suggestive of decline with settlement and, possibly, integration in the host country. As with previous studies, we found that refugees with better German language proficiency had a lower risk of psychological distress, indicating greater integration and economic opportunities [63].

### Limitations and Suggestions for Future Research

This study’s largest limitation lies within the cross-sectional nature of the design, as it can only establish correlational and not causal relationships. A second limitation is that, although the RHS-13 is based on symptoms of multiple mental disorders, including depression, anxiety disorders, and PTSD [47], it is not a diagnostic tool, and does not allow for differentiation between different conditions. Moreover, as a selection bias, reducing the sampling frequency of mentally ill individuals is generally expected in population surveys [64], it is likely that such a bias occurred in the IAB-BAMF-SOEP sampling procedure. Therefore, the rate of psychological distress may have been decreased in the sample. The results of this study concerning regional differences within Germany are only applicable to other countries in a limited manner. Nevertheless, the finding that regional differences are connected to the prevalence of discrimination against refugees and refugee mental health is generally valuable, no matter the country. Additionally, as Germany took in more refugees than any other country within the European Union, during the period of study, the findings are highly relevant.

Moreover, the data collection in 2017 closely followed the increase in attacks against foreigners and refugees across Germany, particularly in eastern Germany, from 2015 to 2016. This proximity in time may have affected the results of this study, especially with regards to perceived discrimination and differences between eastern and western Germany. For this reason, future research should investigate whether the findings of this study can be replicated with data collected both before and after 2015–2016.

## 5. Conclusions

Our findings suggest that, settlement policies of national governments are important in determining the levels of discrimination that refugees are likely to experience, and their consequent distress and mental health problems. Mental health may be supported in urban environments and/or areas where migrants are more likely to be already settled. Female refugees appear to be particularly vulnerable to discrimination and stress, and greater efforts must be made to protect them. Future research may examine whether the reported eastern–western differences hold for the frequency of perceived discrimination, as well as different kinds of discrimination. Such differences between eastern and western Germany likely stem from systematically differing socio-structural factors, like the distribution of wealth, and the number of foreigners living in each part of Germany. More optimistically, there is some evidence that the experience or perception of discrimination may dissolve over time, and with increased language attainment. As the latter factor is protective of mental health, much greater efforts should be made to improve skills at the earliest point after arrival.

## Figures and Tables

**Table 1 ijerph-20-04409-t001:** Sociodemographic characteristics of refugees.

	*n*	%
**Sex**		
Male	1370	66
Female	750	34
**Marital Status**		
Single	715	35
Married/partnership	1351	65
**Country of Origin**		
Afghanistan	253	12
Syria	1081	52
Iraq	249	12
Eritrea	143	7
Others	349	17
**Religion**		
Islam	1425	72
Christian	294	15
No denomination	118	6
Other denomination	150	7
**Level of Education**		
Low	1290	63
Middle	397	19
High	376	18
**Psychological distress**		
Not present	1144	61
present	726	39
**Perceived Discrimination**		
Yes	654	33
No	1359	67
	M	SD
Age	34.55	10.09
German language skills	2.92	0.96

*Note*. Not all characteristic groups add up to N = 2075 due to missing values.

**Table 2 ijerph-20-04409-t002:** Multivariate analysis of psychological distress among refugees settled in Germany (2013–2016).

Odds Ratios (*OR*) with 95% Confidence Intervals (CI)	
**Perceived discrimination**	2.26 (1.80–2.83) **
Eastern Germany	1.34 (1.03–1.74) *
Females	1.68 (1.34–2.10) **
**Age** 18–27 28–34 35–41 42+	1.00 0.89 (0.66–1.19) 0.89 (0.83–1.51) 2.23 (1.64–3.02) **
**Country of origin** Afghanistan Eritrea Iraq Others	1.85 (1.33–2.59) ** 0.70 (0.43–1.15) 1.02 ((0.73–1.42) 1.29 (0.96–1.73)
German language proficiency	1.29 (1.15–1.45) **
Close confidant	1.13 (1.05–1.21) **
Migration traumatic event	1.50 (1.21–1.87) *

*p* value < 0.05 * *p* < 0.005 **. **Reference categories**: no perceived discrimination; western Germany; males; no migration traumatic event.

**Table 3 ijerph-20-04409-t003:** Multivariate analysis of perceived discrimination of refugees settled in Germany (between 2013 and 2016).

Odds Ratios (OR) with 95% Confidence Intervals (CI)	
	OR CI
Eastern Germany	2.53 (1.98–3.22) **
Rural area	1.27 (1.02–1.57) *
Females	0.94 (0.76–1.15)
Age	0.97 (0.96–0.98) **
**Country of Origin** Syria Afghanistan Eritrea Iraq Others	1.00 1.15 (0.84–1.56) 0.88 (0.59–1.31) 0.53 (0.37–0.76) * 1.35 (1.03–1.76) **
Religious Attendance (yes)	1.29 (1.05–1.58) *

*p* value < 0.05 * *p* < 0.005 **. **Reference categories:** western Germany; urban area; males.

## Data Availability

Data on the second wave (2017) of the IAB-BAMF-SOEP survey of refugees in Germany can be supplied on request.

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
