# Peer review of "Geographical Disparities and Settlement Factors and Mental Health of Refugees Living in Germany"

_ijerph, 2023, doi:10.3390/ijerph20054409_

Round 1

Reviewer 1 Report

The article is very well done, but would be enriched by the inclusion of theoretical models that help explain the higher rates of psychological distress among certain categories of migrants and also account for why selected parts of Germany display higher rates of intolerance toward refugees and migrants. This might be a model from theories of discrimination that would help explain why some migrants are at greater risk (women and older refugees) and why some parts of Germany (eastern and rural) are less receptive). The reader is left to intuit an explanatory model.

It may be helpful to discuss the risk and protective factors for psychological distress among refugees and migrants. The authors do observe that close friendships are a protective factor.

Author Response

Reviewer 1

The article is very well done, but would be enriched by the inclusion of theoretical models that help explain the higher rates of psychological distress among certain categories of migrants and also account for why selected parts of Germany display higher rates of intolerance toward refugees and migrants. This might be a model from theories of discrimination that would help explain why some migrants are at greater risk (women and older refugees) and why some parts of Germany (eastern and rural) are less receptive). The reader is left to intuit an explanatory model.

We thank the reviewer for their very supportive comments. On reflection, we could have given more space to a theoretical explanation as to regional and other factors for the differences noted for the discrimination and consequently, psychological distress. We have therefore  provided more text on theoretical aspects of discrimination related to migration histories, level of contact and economic disadvantage. We have also added some text on sex-based differences and duration of stay.

“As in bordering country members such as Hungary, Poland, and the Czech Republic, which all strongly rejected welcome to refugees, Eastern Germans have also been more resistant to European Union ‘open-door’ policies. This resistance has also been expressed in the political system with the far-right Alternative for Germany (AfD) Party winning 13% of the national vote in the 2017 election but recorded double that percentage in Saxony (eastern Germany).  The explanation for higher levels of anti-migrant/refugee sentiment in eastern Germany may be explained by Allport’s contact theory (Allport, 1979) which holds that under certain conditions, stereotyping, prejudice and discrimination could be mitigated by better contact management and greater interpersonal contact between groups.

Logically, the key to this is greater interpersonal contact between groups. However, compared to their Western counterparts, few people growing up in Soviet East Germany prior to unification were able to watch Western television or listen to radio broadcasts, limiting multicultural exposure. Additionally, foreigners in East Germany from Cuba and African countries were mostly visitors rather than settlers, unlike the Turkish in West Germany. Scarce resources, actual or perceived, may also determine hostility to foreigners.  Since unification, the unemployment rate in the East is 13%, remains almost double that in the West (Federal Statistical Office Germany, 2010). “

It may be helpful to discuss the risk and protective factors for psychological distress among refugees and migrants. The authors do observe that close friendships are a protective factor. Thank you for this, we have added further text on this.

Reviewer 2 Report

The paper explores a very interesting, but, at the same, challenging topic. The main aim of this paper is to investigate the effect of discrimination on the mental health of refugees living in Germany and potential eastern-western difference (lines 72-74). The main hypothesis is that the experience of discrimination is positively associated with psychological distress and refugees in eastern Germany would report higher discrimination against compared to western Germany (lines 74-76).

The presentation of the quantitative analysis of the empirical material seems reasonable and straightforward. However, I wonder whether the paper contributes to the discussion on the discrimination of refugees in Germany in particular, and in Europe more generally. Although the paper shows that those refugees who are living in eastern German feel more discriminated than those in Western Germany, there is no justification for this finding. I would expect this finding to be explored further as to elaborate on the variables which might have explanatory value. In section 4.1. the authors mention that they are discussing correlational and not causal relationships, but these correlations need to be more substantial than illustrating profound differences. Therefore, my suggestion is that the data is further elaborated to offer indications of more specific correlations between the available variables. For example, the title refers to settlement factors (!?) which are not clarified in the manuscript. Is there information on more detailed settlement factors, e.g. type of housing (houses, blocks, camps, etc.) for refugees, area/suburb of settlement, rural/urban settlement, etc. This information may offer an elaboration of geographical disparities. Geographical differences are not self-explanatory, as socioeconomic and other variables may intervene (mediate) in the correlation of the findings.

Please note the following critical points which need to be addressed by the authors:

First, the main research question is poorly addressed by the quantitative analysis included in the paper. Further elaboration of the data is strongly recommended in search of the possible factors underlying the difference in discrimination between Western and Eastern Germany and in connection to more specific settlement factors. Clearly, the introduction needs to offer a more extended theoretical discussion on the possible factors addressing the research question (see next points). 

Second, the paper does not seem to make use of a theoretical framework, which might be useful for drawing some more substantive conclusions. By saying theoretical framework, I am not asking for an extensive theoretical discussion, but I would like to see how theory (on the integration of refugees, psychology, migration policy etc.) might inform the discussion of the empirical findings. Part of it could be incorporated in the introductory section and another part in the discussion section. 

Third, the discussion and concluding sections need more substantive elaboration of the findings. The current discussion/conclusion states the obvious and does not make clear why these findings are valuable and in which context. As there are numerous points on men’s and/or women’s religious involvement, I wonder whether those who are more religious feel more threatened compared to the non-religious. Also, feeling threatened may not lead to discrimination. This should be discussed in more detail – as there are some arguments mentioned in lines 277-293.

More to the point, hate speech, xenophobic behaviour (events) and racist incidents could be discussed in relation to the findings. Moreover, I would like to see whether the newcomers (recently arrived, i.e. 6 months) feel more/less discriminated than those who arrived earlier and have better knowledge of German language. The interrelationship of different variables may create b better explanations, arguments or lines of thinking.

Finally, the paper does not make a substantial contribution to the relevant literature. This disadvantage should be remedied in the discussion and concluding sections. In the concluding section I would like to see more on the policy implications of the paper’s findings. Moreover, in the discussion section the findings should be seen in view of the relevant policies (and their implementation). Apparently, the main research question should be fully addressed both in the discussion and concluding sections.

I think the paper needs further work and more substantive arguments to clarify the broad-brush findings which are currently based on expected correlations of the empirical data. More thinking, theoretically informed analysis and justifications are required to ensure that the findings are much better related to the socioeconomic and political setting, while also some explanatory variables are suggested in the end.

Author Response

Reviewer 2

The paper explores a very interesting, but, at the same, challenging topic. The main aim of this paper is to investigate the effect of discrimination on the mental health of refugees living in Germany and potential eastern-western difference (lines 72-74). The main hypothesis is that the experience of discrimination is positively associated with psychological distress and refugees in eastern Germany would report higher discrimination against compared to western Germany (lines 74-76).

We are pleased that the reviewer feels that this is an interesting but challenging topic.

The presentation of the quantitative analysis of the empirical material seems reasonable and straightforward. However, I wonder whether the paper contributes to the discussion on the discrimination of refugees in Germany in particular, and in Europe more generally. Although the paper shows that those refugees who are living in eastern German feel more discriminated than those in Western Germany, there is no justification for this finding. I would expect this finding to be explored further as to elaborate on the variables which might have explanatory value. In section 4.1. the authors mention that they are discussing correlational and not causal relationships, but these correlations need to be more substantial than illustrating profound differences. Therefore, my suggestion is that the data is further elaborated to offer indications of more specific correlations between the available variables. For example, the title refers to settlement factors (!?) which are not clarified in the manuscript. Is there information on more detailed settlement factors, e.g. type of housing (houses, blocks, camps, etc.) for refugees, area/suburb of settlement, rural/urban settlement, etc. This information may offer an elaboration of geographical disparities. Geographical differences are not self-explanatory, as socioeconomic and other variables may intervene (mediate) in the correlation of the findings.

We are not sure what the reviewer means by “no justification for this finding”. We have rigorously examined all of the available data in this major nationally representative dataset. Unfortunately, the dataset does not contain more finely grained variables such as housing type or blocks as the reviewer suggests. However, we completely agree with the reviewer that geography itself does not explain differences. Rather, it is the social and political atmosphere that may determine the reception that migrants and refugees (or just foreigners) experience. The “settlement factors” to which we refer include the region (east/west), experience of discrimination, social support, and language attainment. Their attendance at religious events may also be considered as a settlement factor. Importantly, we did examine urban and rural differences in perceived discrimination and found that rural settlement areas are more likely to be associated with perceived discrimination.

 We also agree with the reviewer that we could have done much more to provide a better theoretical basis for the East/West differences.

Please note the following critical points which need to be addressed by the authors:

First, the main research question is poorly addressed by the quantitative analysis included in the paper. Further elaboration of the data is strongly recommended in search of the possible factors underlying the difference in discrimination between Western and Eastern Germany and in connection to more specific settlement factors. Clearly, the introduction needs to offer a more extended theoretical discussion on the possible factors addressing the research question (see next points). 

Second, the paper does not seem to make use of a theoretical framework, which might be useful for drawing some more substantive conclusions. By saying theoretical framework, I am not asking for an extensive theoretical discussion, but I would like to see how theory (on the integration of refugees, psychology, migration policy etc.) might inform the discussion of the empirical findings. Part of it could be incorporated in the introductory section and another part in the discussion section. 

We agree that the theoretical basis for east/west refugee experiences requires more elaboration and have substantially added to the introduction and discussion.

Third, the discussion and concluding sections need more substantive elaboration of the findings. The current discussion/conclusion states the obvious and does not make clear why these findings are valuable and in which context. As there are numerous points on men’s and/or women’s religious involvement, I wonder whether those who are more religious feel more threatened compared to the non-religious. Also, feeling threatened may not lead to discrimination. This should be discussed in more detail – as there are some arguments mentioned in lines 277-293. We have tried to better convey in the paper that the data can only provide the participants’ reported experiences of discrimination – that is, they are uncorroborated accounts of discrimination. Nevertheless, there is evidence that refugees have considerable experience of hostility and discrimination and that these may be more prevalent in eastern Germany than western Germany.  These points have been addressed in the discussion

More to the point, hate speech, xenophobic behaviour (events) and racist incidents could be discussed in relation to the findings. Moreover, I would like to see whether the newcomers (recently arrived, i.e. 6 months) feel more/less discriminated than those who arrived earlier and have better knowledge of German language. The interrelationship of different variables may create b better explanations, arguments, or lines of thinking. Unfortunately, the database does not contain variables on hate speech, but we have provided data on perceived discrimination and its impact on mental health, an effect which has been shown in other studies.  It is important to stress that we did however, include language attainment and age in a logistic regression analysis of independently associated factors.As the reviewer notes, language attainment is strongly and independently associated with the outcomes. We also noted that perceived discrimination decreases with age. While we cannot firmly conclude that this equates with length of stay, it is certainly suggestive of decline with settlement and possibly, integration in the host country. We have taken the reviewer’s suggestion to highlight this more nuanced finding.

Finally, the paper does not make a substantial contribution to the relevant literature. This disadvantage should be remedied in the discussion and concluding sections. In the concluding section I would like to see more on the policy implications of the paper’s findings. Moreover, in the discussion section the findings should be seen in view of the relevant policies (and their implementation). Apparently, the main research question should be fully addressed both in the discussion and concluding sections. We thank the reviewer and have added more discussion on policy implications.

I think the paper needs further work and more substantive arguments to clarify the broad-brush findings which are currently based on expected correlations of the empirical data. More thinking, theoretically informed analysis and justifications are required to ensure that the findings are much better related to the socioeconomic and political setting, while also some explanatory variables are suggested in the end. Thanks, we have added a more theoretical basis and contextual, explanatory material (see below, for example). We hope this allows a better basis for understanding the results.

As in bordering country members such as Hungary, Poland, and the Czech Republic, which all strongly rejected welcome to refugees, Eastern Germans have also been more resistant to European Union ‘open-door’ policies. This resistance has also been expressed in the political system with the far-right Alternative for Germany (AfD) Party winning 13% of the national vote in the 2017 election but recorded double that percentage in Saxony (eastern Germany).  The explanation for higher levels of anti-migrant/refugee sentiment in eastern Germany may be explained by Allport’s contact theory (Allport, 1979) which holds that under certain conditions, stereotyping, prejudice and discrimination could be mitigated by better contact management and greater interpersonal contact between groups. Logically, the key to this is greater interpersonal contact between groups. However, compared to their Western counterparts, few people growing up in Soviet East Germany prior to unification were able to watch Western television or listen to radio broadcasts, limiting multicultural exposure. Additionally, foreigners in East Germany from Cuba and African countries were mostly visitors rather than settlers, unlike the Turkish in West Germany. Scarce resources, actual or perceived, may also determine hostility to foreigners.  Since unification, the unemployment rate in the East is 13%, remains almost double that in the West (Federal Statistical Office Germany, 2010).
